# Congenital Heart Disease from Infancy to Adulthood: Pathology and Nosology

**DOI:** 10.3390/biomedicines13040875

**Published:** 2025-04-04

**Authors:** Gaetano Thiene, Marny Fedrigo

**Affiliations:** Cardiovascular Pathology, Department of Cardiac, Thoracic, Vascular Sciences and Public Health, Medical School, University of Padua, 35123 Padova, Italy; marny.fedrigo@gmail.com

**Keywords:** adulthood, bicuspid aortic valve, cardiac surgery, congenital heart diseases, elderly, molecular pathology

## Abstract

Congenital heart diseases (CHDs) are usually defined as structural anomalies of the heart and great arteries, present since birth, that are due to embryological maldevelopment, with overt or potential dysfunction. Nowadays, most of the patients with CHD in adulthood (age > 18 years) had been operated on with success in infancy or childhood and undergo periodical screening. Pathology and nosology of CHDs are herein treated with special attention to adulthood according to the involved cardiac structures (aorta, valves, coronary arteries, myocardium, great arteries, conduction system). Moreover, the purpose is to postulate, in the era of molecular medicine, that genetically determined defects are also congenital cardiac disorders, with or without structural abnormality, and should be defined CHDs as well since their molecular background is material and present since conception.

## 1. Introduction

Traditionally, congenital heart diseases (CHDs) are defined as structural anomalies of the heart and great arteries, present at birth, that are due to embryological maldevelopment, with evident or potential dysfunction [1,2,3,4,5,6,7,8,9,10,11,12,13].

Nowadays, almost all CHDs may be repaired with success in infancy and childhood, requiring periodic check-up. Actually, they represent the majority of patients with CHD in adulthood (>18 years old), who may achieve a life expectancy almost equal to subjects born with normal hearts [11]. Moreover, many structural CHDs may escape diagnosis at birth and remain silent until adulthood. Genetically determined cardiac diseases, whether structural or functional, should be considered CHDs as well.

We approach the topic on the basis of the involved cardiac structures: coronary arteries, great arteries, myocardium, valves, and conduction system.

## 2. Coronary Arteries

The origin of an epicardial coronary artery from the pulmonary artery is usually lethal at birth, incompatible with life, and exceptionally seen in people growing up [14].

The origin of a coronary artery from the wrong aortic sinus (Figure 1A) is one of the most frequent causes of sudden death in the young, particularly in grown-up athletes (>18 years old) [15], precipitated by a strong effort. It is at risk of myocardial ischemia, because the first tract runs between the aorta and pulmonary artery, with the lumen flattened during systolic blood ejection. Moreover, the initial course may be intramural of the aortic wall (Figure 1B). High take-off from the aorta and the retroaortic course of the circumflex artery arising from the right coronary artery may also be life threatening, even late in adulthood, because of myocardial ischemia [15].

A controversial issue is the course of the descending coronary artery within the myocardium (Figure 2). It is so frequent as to be considered a variant of normal. The myocardium, covering the coronary segment, is known as a “bridge” (Figure 2A). It is a risky abnormality when the course is intramural, long and deep and when surrounded by a sleeve of myocardium prone to spasm (Figure 2C) [15]. Concealed coronary atherosclerosis may appear in childhood and become worrisome at early adulthood (Figure 3) [15].

## 3. Aorta and Pulmonary Artery

A rare CHD, reaching even the elderly, is corrected transposition of the great arteries (Figure 4A). The term “corrected” means “physiologically corrected” by atrioventricular and ventriculoarterial discordance in sequence, ensuring normal physiology of the blood stream (systemic and pulmonary circulations in series and not parallel, as in complete transposition) [16]. In this CHD, the AV conduction is located anteriorly with progressive fibrotic degeneration of the His bundle, putting the patient at risk of sudden death by AV block [17] (Figure 4B). Oddly enough, complete transposition may also exceptionally reach adulthood when associated with pulmonary stenosis and a ventricular septal defect.

Another paucisymptomatic CHD is coarctation of the aortic arch (Figure 5), with ischemia of inferior limbs. It is at risk of lethal complications by aortic dissection because of proximal arterial hypertension and obstructive coronary atherosclerosis (Figure 6), which may trigger myocardial ischemia, myocardial infarction, and even sudden death. Moreover, acquired aneurysms of cerebral arteries in the Willis circuit can undergo rupture and fatal subarachnoid hemorrhage.

## 4. Valves

The bicuspid aortic valve (BAV), the most frequent CHD, is not an innocent malformation, because of the occurrence of progressive dystrophic calcification with early aortic stenosis (Figure 7) and incompetence by dilatation of the sinusal ascending aorta, ascribable to aortopathy by Erdheim’s medionecrosis (Figure 8), at risk of dissection in the grown-up patient (Figure 9). The BAV syndrome may be inherited, requiring screening of the family for primary prevention [18]. Even the pulmonary valve may show an anomalous number of cusps, late observed in the elderly (Figure 10).

Mitral valve prolapse is a CHD because of the disjunction of the atrial and ventricular myocardium at the annulus insertion (Figure 11) [19]. In this setting, the billions of ventricular systoles which close the valve orifice during life account for traumatic injury of the leaflets with myxoid degeneration, mitral prolapse, and incompetence, as well as fibrosis of the papillary muscles and ventricular myocardium, which is an arrhythmic substrate placing the patient at risk of sudden death [19].

The Ebstein anomaly [20,21,22], with lowering of the leaflets and incompetence of the tricuspid valve (Figure 12 and Figure 13), is associated with myocardial dystrophy of the right ventricle free wall (Figure 14) and with ventricular preexcitation (Figure 15).

## 5. Myocardium

The noncompacted left ventricle (Figure 16) is a CHD due to embryological failure of ventricular myocardium compaction. It accounts for progressive ventricular dilatation and severe congestive heart failure, mimicking dilated cardiomyopathy because of the scarcity of contractile myocardium. The fissures in between the trabeculae are so deep that the endocardium may reach the epicardium. Moreover, the trabeculae are so big as to be erroneously interpreted as mural thrombi of a dilated cardiomyopathy by 2D echo.

## 6. Septal Defects

Septal defects with a left to right shunt, like an atrial septal defect, ventricular septal defect, AV canal, truncus arteriosus, aorto-pulmonary window, and patent ductus arteriosus, may persist until adulthood if not diagnosed and repaired early by surgery or interventional cardiology. They can be complicated with pulmonary hypertension by pulmonary vascular diseases (Figure 17), with the appearance of cyanosis because of an inverted shunt from right to left, as in Eisenmenger syndrome (Figure 18) [23,24].

## 7. Conduction System

Concerning CHDs of the conduction system in adulthood, ventricular preexcitation (VPE) represents the smallest malformation, consisting of a fascicle of ordinary myocardium (1–2 mm thick) known as the Kent fascicle. It is usually located in the lateral rings (Figure 19A). It connects the atria to the ventricles (Figure 19B) and is bare of the decremental properties of the Tawarian System, which the opposite, as it is located in the central AV septum and consists of specialized conducting tissue endowed with decremental properties.

The Kent fascicle accounts not only for VPE but also for atrial reentry with palpitations by supraventricular tachycardia. In the case of an atrial fibrillation burst (usually about 500 beats per minute), because of the absence of a slowing-down of the electrical impulse, a step one-to-one transmission to the ventricles may trigger a lethal ventricular fibrillation (Figure 20).

Also, progressive AV block (Figure 21), usually manifesting in in elderly patients, may be genetically determined due to a mutation of the SN5 (Sodium channel five) gene.

## 8. Should Genetically Determined Cardiovascular Diseases Be Considered Congenital Heart Diseases? A Nosological Problem

Different from most of the so-called CHDs (90% due to embryological maldevelopment) are cardiac and vascular diseases that are genetically determined in the code of life.

The genetic defect of channelopathies, like simple missense mutation, lies in the double helix of the DNA and is usually transmitted at the time of conception but may manifest phenotypically with growing up. Should channelopathies be termed CHDs? Should genetically determined cardiomyopathies (like hypertrophic, dilated, restrictive, and arrhythmogenic ones) be considered CHDs as well? Why not?

Fibroelastosis is the consequence of a viral infection (mumps) during fetal life. Is it a CHD?

As for birth defects, Wikipedia states, “A birth defect is an abnormal condition that is present at birth, regardless of its cause. [...] Birth defects are divided into two main types: structural disorders, in which problems are seen with the shape of a body part, and functional disorders in which problems exist with how a body part works. Some birth defects include both structural and functional disorders”. Congenital functional disorders may be genetically determined and as such they are CHDs.

Genetic defects of DNA may manifest after infancy or childhood. Is a wrong molecular sequence of DNA (even a small missense mutation) enough to be called a CHD? Are babies with a channelopathies-like long QT affected by a CHD? (See Figure 22). The substrate of channelopathies is molecular and not anatomic–structural, as seen earlier in the definition of CHDs. However, molecular genes code wrong amino acids and proteins, which are material rather than dysfunctional.

Another example is Pompe disease (glycogenosis with a big heart at birth by intracellular glycogen storage by gene defect) (Figure 23). Why is this not a CHD? It is clearly a structural defect present at birth, in agreement with the actual CHD definition.

As for sudden death in the young, there are many cases of “mors sine materia”, namely, with an apparently normal heart without a structural defect. For instance, molecular investigation finds the cause of sudden death in catecholaminergic tachycardia, with mutation of ryanodine [15] or calsequestrin genes, devoted to the electromechanical association of Ca++ release by the smooth reticulum (Figure 24). A missense mutation can cause sudden arrhythmic death without an anatomical substrate [15]. Why is this not considered sudden death due CHD?

## 9. Conclusions

CHDs are usually defined as structural anomalies of the heart. The success of cardiac surgery is improving longevity and the burden of adult CHDs. There are several cardiac disorders that are molecular, with dysfunction of the electrical activity of the heart.

They deserve to be considered congenital heart disease even if the defect is molecular, at the level of the code of life.

## Figures and Tables

**Figure 1 biomedicines-13-00875-f001:**
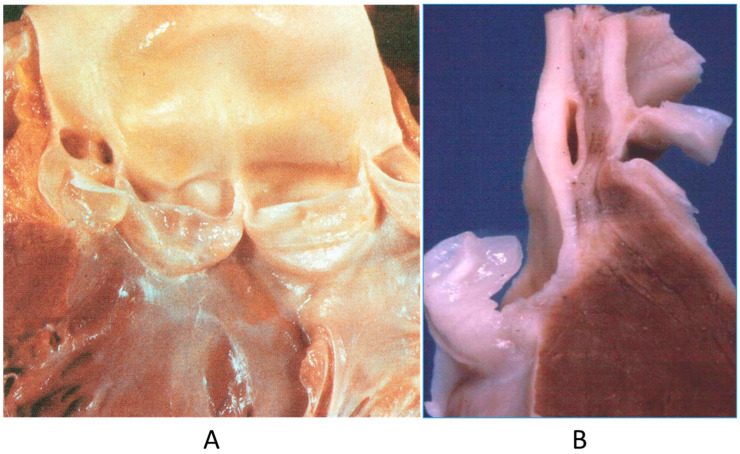
(**A**) Both coronary arteries take the origin from the anterior left sinus of Valsalva, from a 22-year-old male who died suddenly during a soccer game. (**B**) The intramural aortic course of the anomalous artery [15].

**Figure 2 biomedicines-13-00875-f002:**
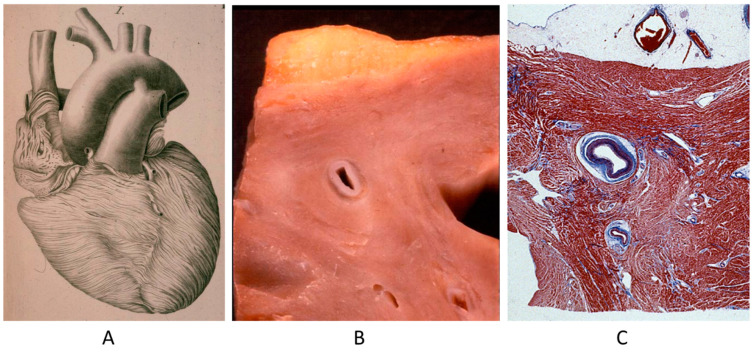
Anomalous intramural myocardial course of the descending coronary artery. (**A**) Myocardial bridge. (**B**) Intramural course. (**C**) Sleeve of myocardium around the intramural coronary artery [15].

**Figure 3 biomedicines-13-00875-f003:**
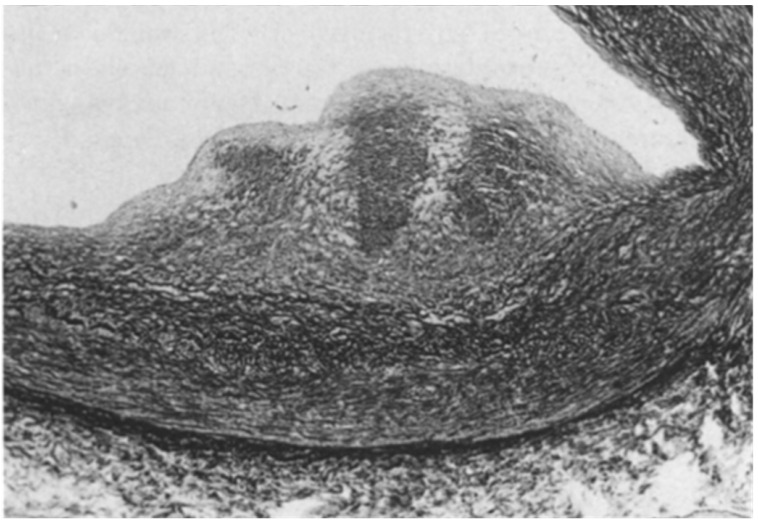
Coronary artery atherosclerotic plaque in a 19-year-old boy.

**Figure 4 biomedicines-13-00875-f004:**
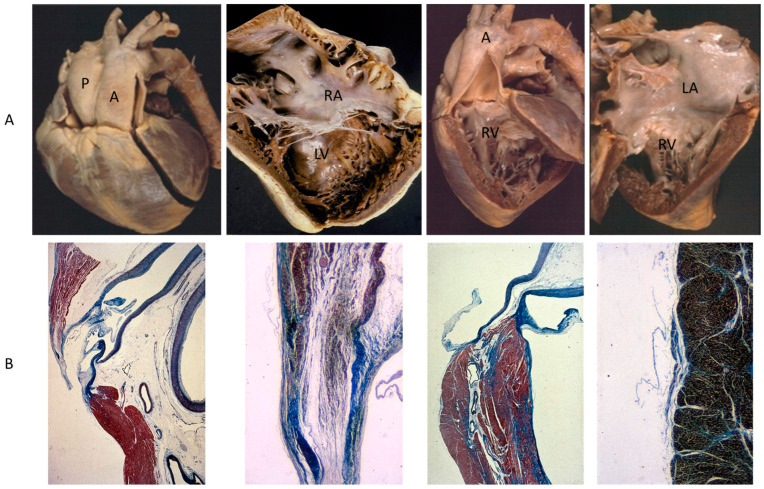
(**A**) Corrected transposition with atrioventricular and ventricular arterial discordance. (**B**) The AV node is anteriorly located and the His bundle interrupted by fibrosis. From a 43-year-old female with sudden death by AV block. A = aorta; LA = morphologically left atrium; LV = morphologically left ventricle; P = pulmonary artery; RV = morphologically right ventricle, RA= morphologically right atrium.

**Figure 5 biomedicines-13-00875-f005:**
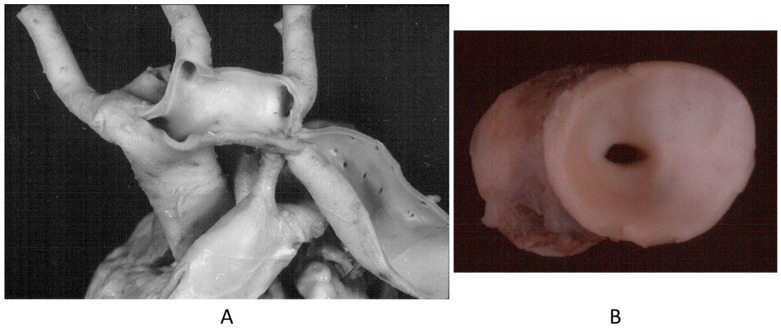
Aortic coarctation (**A**,**B**) in adulthood.

**Figure 6 biomedicines-13-00875-f006:**
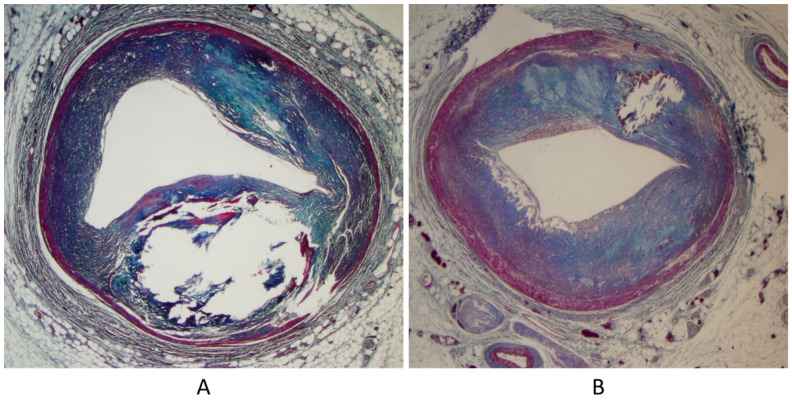
Sudden death in aortic coarctation by obstructive coronary atherosclerosis (**A**,**B**).

**Figure 7 biomedicines-13-00875-f007:**
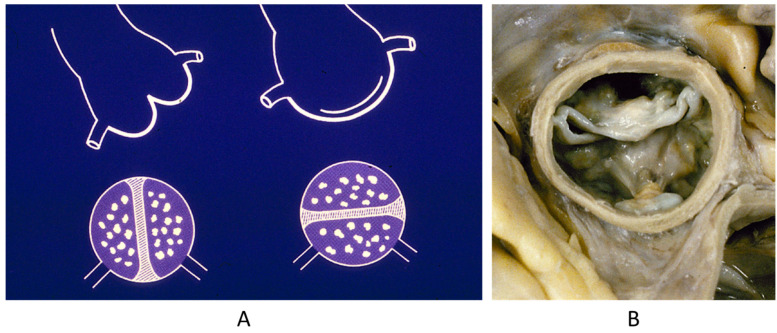
Bicuspid aortic valve (**A**) with severe calcific stenosis (**B**) in a 65-year-old man [18].

**Figure 8 biomedicines-13-00875-f008:**
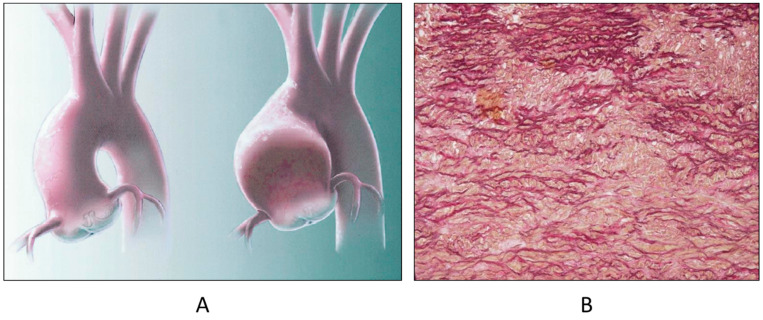
Aortopathy associated with bicuspid valve with dilatation of sinusal ascending aorta (**A**) and elastic lamellae disruption in the aortic tunica media (**B**) [18].

**Figure 9 biomedicines-13-00875-f009:**
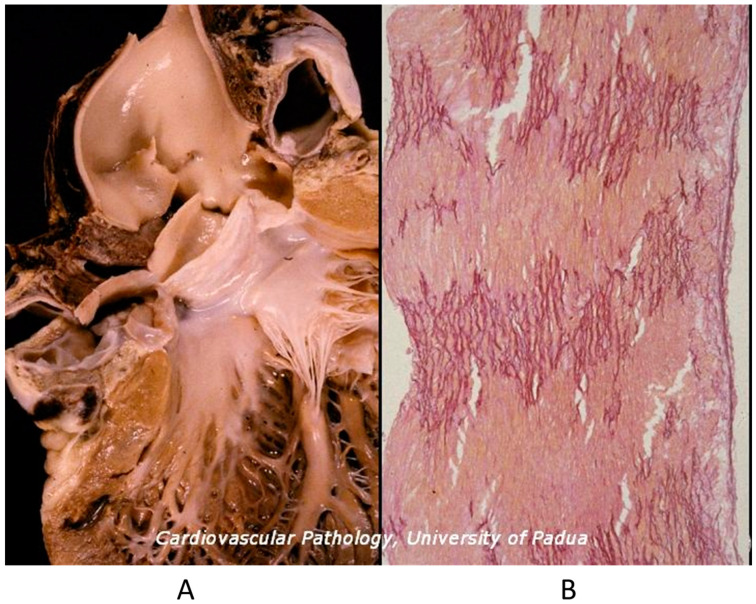
Fatal dissecting aneurysm of the ascending aorta in bicuspid aortic valve (**A**) with disappearance of elastic lamellae in the tunica media (**B**) [18].

**Figure 10 biomedicines-13-00875-f010:**
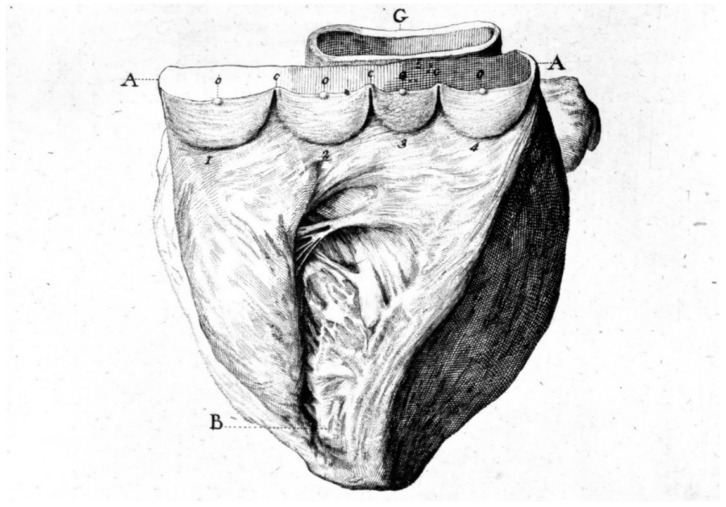
A quadricuspid pulmonary valve incidentally observed in a 70-year-old patient. A = quadricuspid pulmonary valve, B = right ventricle, C = Aorta.

**Figure 11 biomedicines-13-00875-f011:**
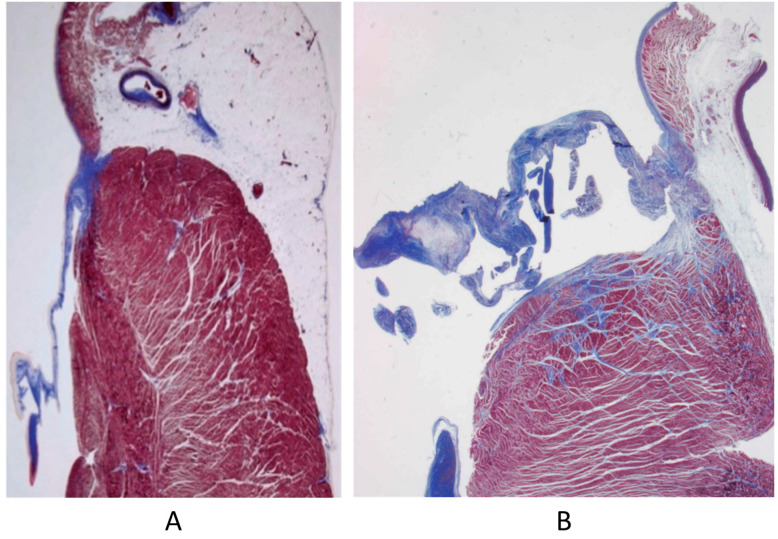
Normal mitral annulus (**A**). Mitral annulus with annular disjunction (**B**) [19].

**Figure 12 biomedicines-13-00875-f012:**
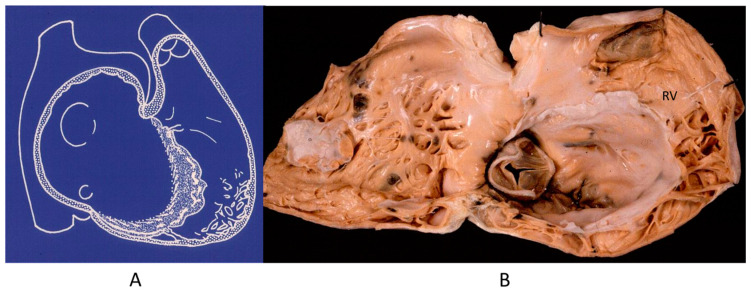
Ebstein anomaly with downward displacement of the tricuspid valve leaflets (**A**). Note a porcine bioprosthetic valve in place of the tricuspid valve (**B**) [9]. RV = right ventricle [20].

**Figure 13 biomedicines-13-00875-f013:**
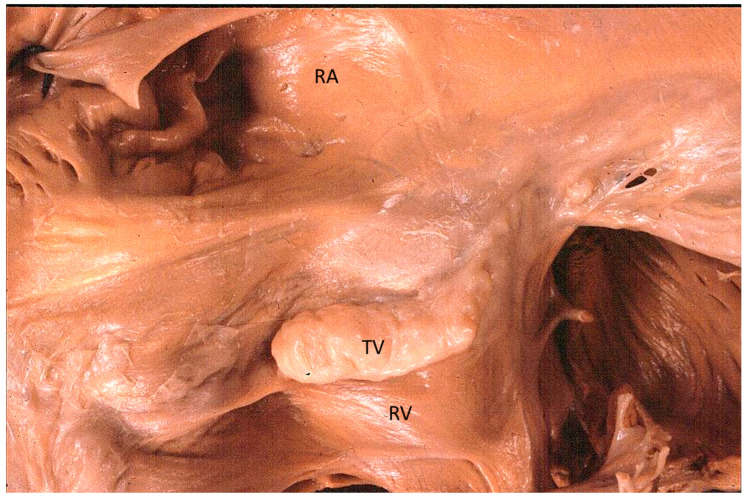
Other examples of the Ebstein anomaly, with the septal leaflet of the tricuspid valve downward displaced in an adult woman [20]. RA = right atrium; RV = right ventricle; TV = tricuspid valve [20].

**Figure 14 biomedicines-13-00875-f014:**
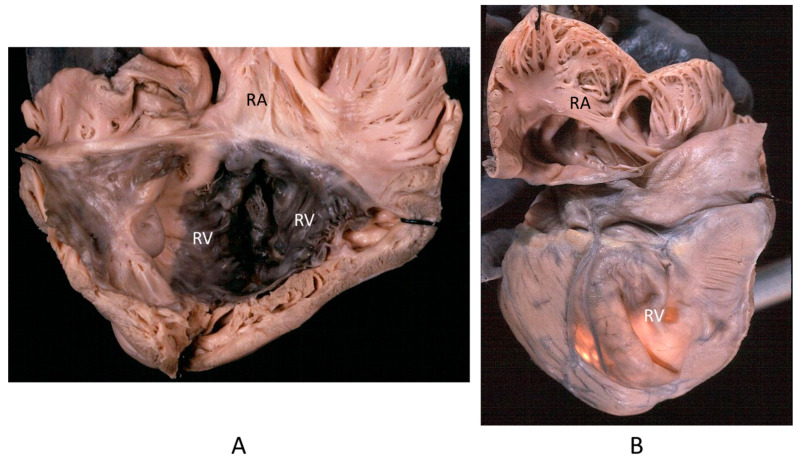
Aneurysm by dystrophy of the right ventricular wall (**B**) in an adult patient with Ebstein anomaly of the tricuspid valve (**A**). RA = right atrium; RV = right ventricle.

**Figure 15 biomedicines-13-00875-f015:**
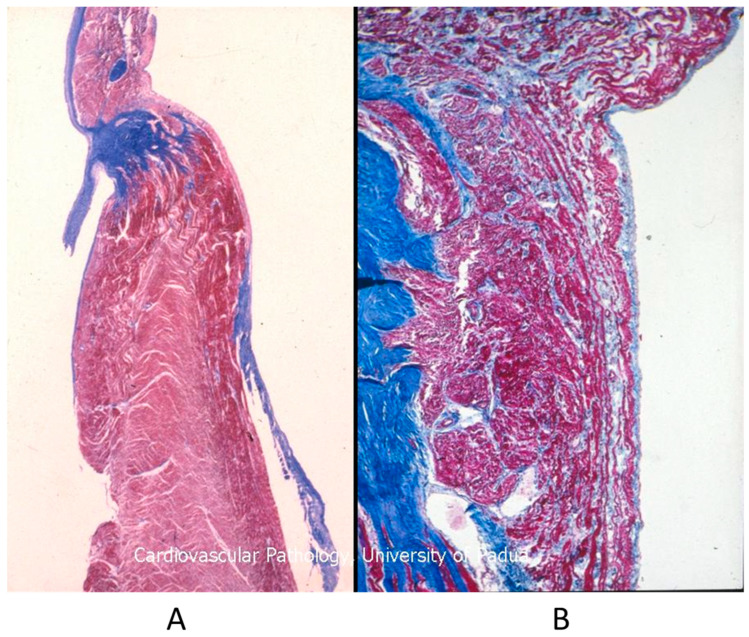
Histology of Ebstein anomaly with displaced septal leaflet of the tricuspid valve (**A**) and Kent septal fascicle (**B**) in a young person who died suddenly [20].

**Figure 16 biomedicines-13-00875-f016:**
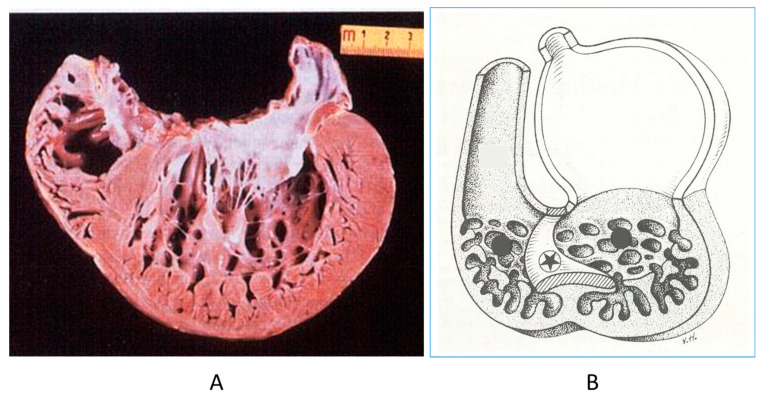
A gross view of a heart specimen with biventricular noncompaction (**A**) and drawing of noncompacted ventricles (**B**).

**Figure 17 biomedicines-13-00875-f017:**
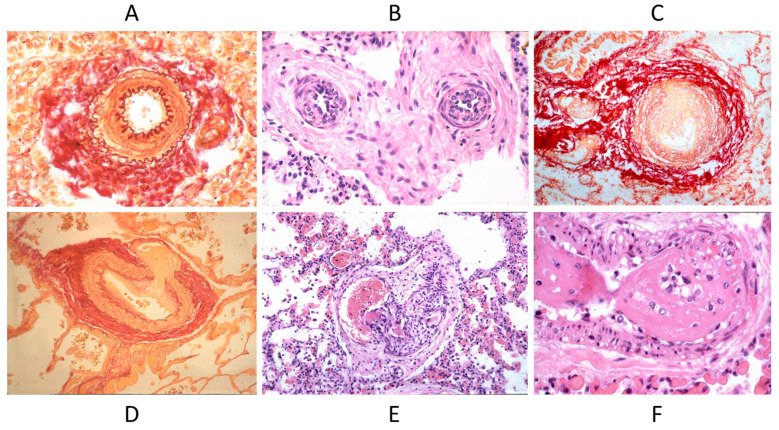
Pulmonary vascular disease in Eisenmenger syndrome with obstructed lumen (**A**–**C**) and aneurysm (**D**), glomerular-like proliferation (**E**), and fibrinoid necrosis (**F**).

**Figure 18 biomedicines-13-00875-f018:**
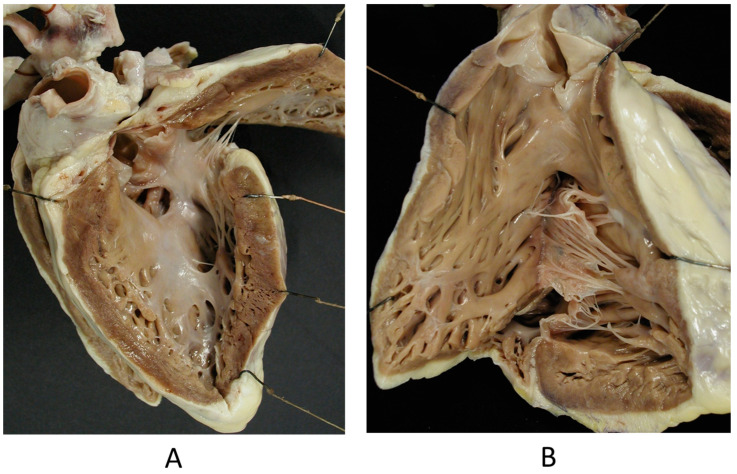
Eisenmenger complex in 20-year-old boy with biventricular origin of the aorta (**A**), without pulmonary stenosis (**B**).

**Figure 19 biomedicines-13-00875-f019:**
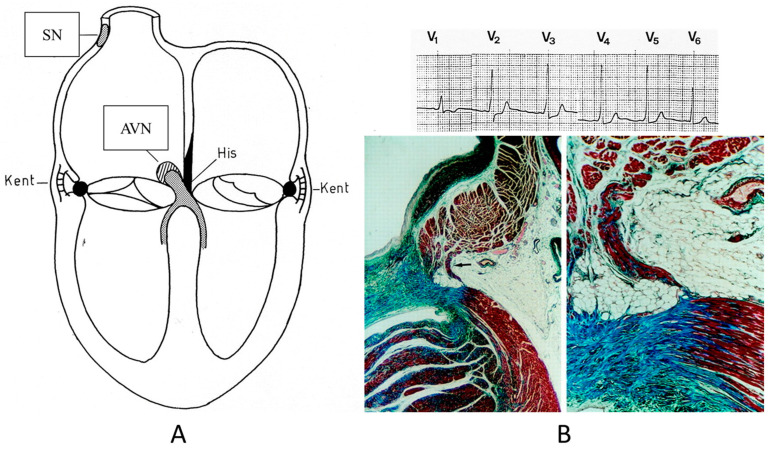
Schematic representation of ventricular preexcitation with Kent fascicle in the lateral atrio-ventricular rings (**A**). Histology of Kent fascicle and electrocardiogram with delta wave (**B**). SN = sinus node, AVN = atrio-ventricular node.

**Figure 20 biomedicines-13-00875-f020:**
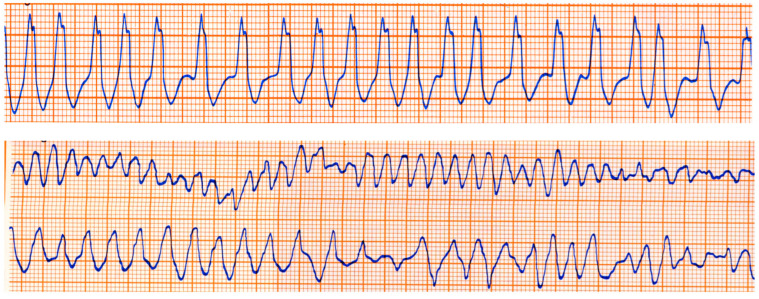
Sudden death in Wolff Parkinson White syndrome: atrial fibrillation turns into ventricular fibrillation.

**Figure 21 biomedicines-13-00875-f021:**
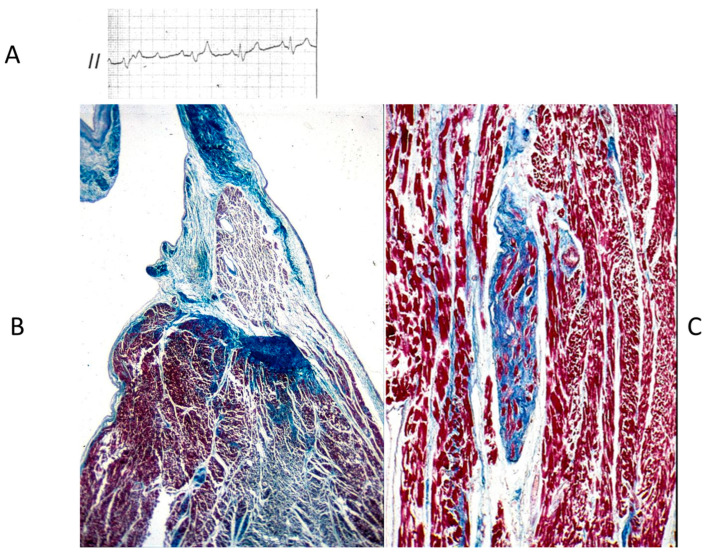
Electrocardiogram with AV block (**A**). At histology, bifurcation of His bundle with fibrous interruption of the left bundle branch (**B**) and the right bundle branch (**C**) were observed [15].

**Figure 22 biomedicines-13-00875-f022:**
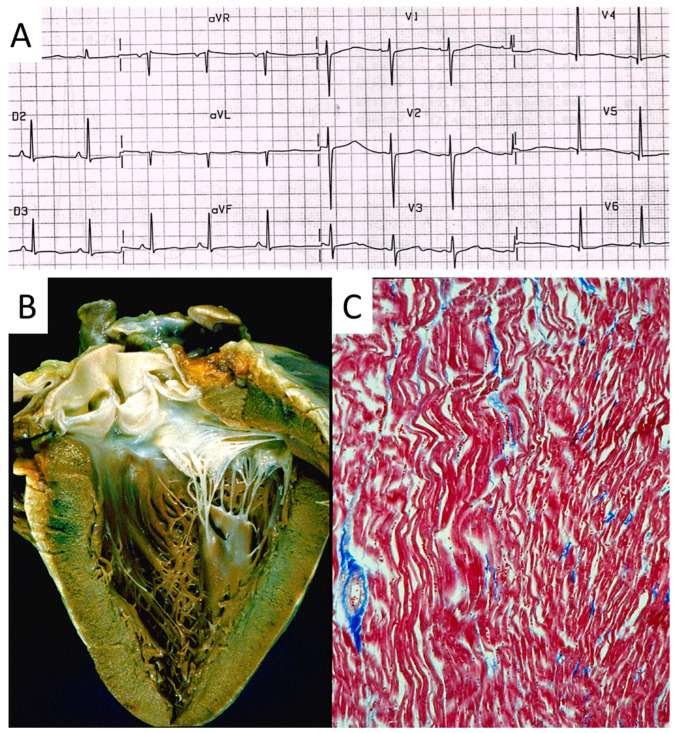
Sudden death in a 20-year-old young adult with long QT in an electrocardiogram image (**A**). The heart is normal in gross (**B**) and histology (**C**) views [15].

**Figure 23 biomedicines-13-00875-f023:**
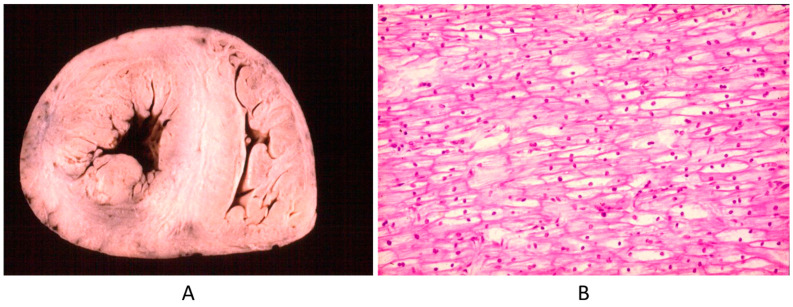
Big heart (**A**) by glycogenosis at histology (**B**) (Pompe disease) in an infant.

**Figure 24 biomedicines-13-00875-f024:**
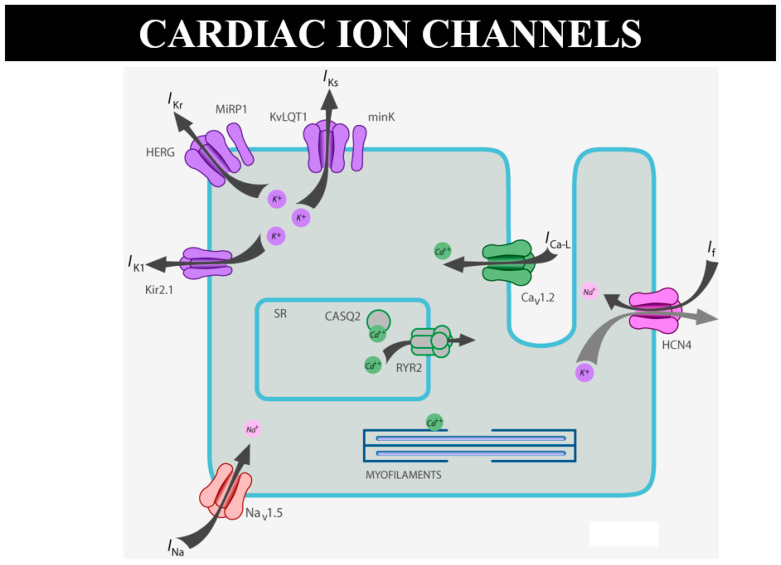
Schematic representation of ion channels and ryanodine receptor, the latter in charge of Ca++ release from the smooth reticulum for electromechanical coupling.

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
