# Peer review of "Congenital Heart Disease from Infancy to Adulthood: Pathology and Nosology"

_biomedicines, 2025, doi:10.3390/biomedicines13040875_

Round 1
Reviewer 1 Report
Comments and Suggestions for Authors
I thank the editor for the opportunity to review this manuscript.
The authors present a manuscript focused on several congenital defects that may manifest in adulthood. However, the manuscript contains multiple shortcomings and, in its current form, is not suitable for publication.
-
The manuscript arbitrarily selects nosological entities from different categories of congenital developmental defects without providing sufficient detail on any of them. This may mislead readers when assessing the risk of individual malformations for affected patients.
-
The scope of the sections is excessively broad for a review article. It is not feasible for a review to comprehensively and clearly capture the content of such an extensive topic.
-
The figure descriptions are unclear in several instances, making it difficult for readers to determine which subfigure (a, b, c) corresponds to a particular explanation.
-
Citations are not provided in the text.
-
For individual nosological entities, the authors frequently omit essential parameters such as etiology, symptomatology, diagnostics, therapy, and treatment efficacy.
In its current form, I do not recommend the manuscript for publication. The authors present highly interesting photographic documentation, which is of significant value to clinical readers. I suggest restructuring the work into multiple review articles, each selectively dedicated to specific subchapters. The authors should provide a more detailed and precise discussion of each diagnosis.
Author Response
The manuscript is a perspective article, not a review paper.
We quoted citations in the text with number. Moreover, we would like to stress that the paper is not clinical with nosological considerations on Congenital Heart Disease definition. Does the reviewer agree with our nosological view point?
Reviewer 2 Report
Comments and Suggestions for Authors
This is a well-illustrated review, or, better, essay on several pathologies, namely anomalous origin of coronary arteries, myocardial bridges, corrected transposition of the great arteries, coarctation of the aortic arch, bicuspid aortic valve, mitral valve prolapse, Ebstein anomaly, non compacted left ventricle, septal defects, WPW, AV block and channelopathy. Thus, mainly congenital diseases that are rarely covered in CHD literature or known as classical CHD (with exception on septal defects and transposition). The paper was not prepared in classical way as a review article, but could serve as editorial to Special issue of the Journal. Otherwise, the paper requires substantial corrections, including data on epidemiology, taxonomy, natural course , pathology features etc.
- The title doesn’t presents the topic of the review. Has to be clarified
- Abstract is brief and universal. , however acceptable and represented the main body of the text.
- The main body of the text consists with short abstracts about several pathology. Good for editorial article , but not for review.
- Conclusion is essential and should be included
- Illustrations are of great value. Especially , historical ones. Moreover, the main hard point of the paper that makes it interesting to readers in current form are illustrations.
- References are updated
Author Response
We thank for the appreciation of the illustrations. We added conclusions.
Reviewer 3 Report
Comments and Suggestions for Authors
The authors give a nice overview of the different congenital anomalies in cardiology. The comments I have are meant to strengthen the manuscript:
- Please address the references in manuscript as appropriate. Number them and cite them in the text. It is difficult for me to check if the appropriate references are used and to which specific manuscript you're referencing to.
- Additionally, please use adequate references for the statements that are being made. For example, I could only find one reference for aortic coarctation which was a case report. There are large cohort studies performed which would be much more appropriate to cite (PMID: 34755532)
Author Response
Thank you for the flattering comments. We numbered the references in the text, when quoting. As far as valves, we added our recent book as reference (8).
Round 2
Reviewer 1 Report
Comments and Suggestions for Authors
Thank you to the editor for the opportunity to review the manuscript once again.
However, I must conclude that the authors have insufficiently addressed the required revisions in the manuscript. Aside from the overall lack of conceptual coherence in the document, several other requested changes have not been adequately implemented, such as the description of the figures. Some figures still lack descriptions for the A and B subsegments.
I recommend rejecting the manuscript for publication in its current form.
Author Response
- We took note that the English is fine and does not require any improvement.
- All the figures show now A and B or more in subsegments.
Reviewer 2 Report
Comments and Suggestions for Authors
As I wrote before, the paper is essential editorial. Otherwise, it is unacceplable
Author Response
We took note that the English is fine and does not require any improvement. The manuscript is not an Editorial, it is a perspective paper.